# Investigation of Electrical Stability and Sensitivity of Electric Double Layer Gated Field-Effect Transistors (FETs) for miRNA Detection

**DOI:** 10.3390/s19071484

**Published:** 2019-03-27

**Authors:** Wen-Che Kuo, Indu Sarangadharan, Anil Kumar Pulikkathodi, Po-Hsuan Chen, Shin-Li Wang, Chang-Run Wu, Yu-Lin Wang

**Affiliations:** 1Institute of Nanoengineering and Microsystems, National Tsing Hua University, Hsinchu 300, Taiwan; jp65k6123@gmail.com (W.-C.K.); indu.4391@gmail.com (I.S.); anilpnarayan@gmail.com (A.K.P.); sam76227@gmail.com (P.-H.C.); w0970711363@gmail.com (S.-L.W.); kl2846@gmail.com (C.-R.W.); 2Department of Power Mechanical Engineering, National Tsing Hua University, Hsinchu 300, Taiwan

**Keywords:** field-effect transistors (FETs), sensors, stability, miRNA, electric double layer (EDL)

## Abstract

In this research, we developed a miRNA sensor using an electrical double layer (EDL) gated field-effect transistor (FET)-based biosensor with enhanced sensitivity and stability. We conducted an in-depth investigation of the mechanisms that give rise to fluctuations in the electrical signal, affecting the stability and sensitivity of the miRNA sensor. Firstly, surface characteristics were studied by examining the metal electrodes deposited using different metal deposition techniques. The lower surface roughness of the gold electrode improved the electrical current stability. The temperature and viscosity of the sample solution were proven to affect the electrical stability, which was attributed to reducing the effect of Brownian motion. Therefore, by controlling the test conditions, such as temperature and sample viscosity, and the surface characteristics of the metal electrodes, we can enhance the stability of the sensor. Metal electrodes deposited via sputtering and e-beam evaporator yielded the lowest signal fluctuation. When ambient temperature was reduced to 3 °C, the sensor had better noise characteristics compared to room temperature testing. Higher viscosity of samples resulted in lower signal fluctuations. Lastly, surface functionalization was demonstrated to be a critical factor in enhancing the stability and sensitivity. MiRNA sensors with higher surface ratios of immobilized DNA probes performed with higher sensitivity and stability. This study reveals methods to improve the characteristics of EDL FET biosensors to facilitate practical implementation in clinical applications.

## 1. Introduction

Field effect transistors (FET) have been used for various sensing applications, such as ion [1,2], gas [3,4], and biomolecule sensing [5,6]. Studies report high sensitivities and other attractive features, such as speedy response, low cost, and ease of use and implementation [7,8]. FET sensors have demonstrated promising potential in the detection of biomolecules [9,10,11]. The superior electrical properties of FET can facilitate the transduction of very minute surface potential changes resulting from surface bound receptors and target biomolecules in the testing aqueous sample. However, in the past, the severe charge screening effect in test environments with very high salt concentrations has been a limitation of FET based biosensors [12]. Device operation in physiological fluids such as serum, blood, or urine required extensive pre-processing of the testing liquid sample, in an effort to reduce the ionic strength of the sample while maintaining the integrity of the target biomolecule. However, this requires complex actuation mechanisms and automation if FET biosensors are to be deployed for non-laboratory based applications. Some other studies have aimed to overcome the Debye screening limitations in high ionic strength media [13,14,15]. They have made use of polymer-based surface functionalization strategies to effectively increase the Debye length and facilitate protein detection, with the polymer acting as an additional capacitive layer. In another study, Donnan’s equilibrium was cited as the physical mechanism behind the elongation of Debye length, which is also effectively capacitance based signal transduction [16]. However, adding more dielectric layers may compromise the sensitivity, and extremely low concentrations of biomolecules may not be detected. The sensor response time was much longer, probably owing to the additional dielectric layers on the sensor surface [13,14,15], and has yet to be proven in clinical samples. In our previous works, we developed high field modulated FET biosensors that featured a unique gating mechanism, which employs the testing liquid sample to act as an additional dielectric in the sensor system [17]. By doing so, we could overcome the limitations of the screening effect and detect biomolecules in their native fluid compositions. With this methodology, we were able to detect proteins, nucleotides, and cells in their native composition [18,19,20,21]. We also implemented an extended gate FET biosensor design, using the same gating mechanism to detect proteins in whole blood [22,23]. The electrical stability and the effects of surface functionalization on the sensor performance needs to be evaluated in a detailed study. 

In this research, we investigated the electrical stability of the high field modulated extended gate FET biosensor in aqueous test environments. The reported high sensitivities of the FET biosensor indicate that slight variations in surface potential can result in electrical signal change, in which case, maintaining electrical stability during sensor operation assumes paramount importance. We have investigated different types of sensor surfaces and their influence on the electrical characteristics of the FET biosensor. By varying the ambient temperature and test sample’s viscosity, we have demonstrated that Brownian motion is highly relevant to maintaining electrical stability. One of the most critical factors in affinity sensors, surface functionalization, has also been shown to enhance or reduce electrical stability based on the density of the surface bound receptors. This study thus guides us to develop an optimized sensor and experimental design that facilitates stable and highly sensitive biomolecule detection in physiological conditions. 

## 2. Experimental Section

### 2.1. Sensor Chip Fabrication

The sensor chip containing the gold electrodes was fabricated on a polymer substrate. A master mold of poly-methyl methacrylate (PMMA) was used to fabricate a Polydimethylsiloxane (PDMS) mold. The PDMS poured into the PMMA mold was cured for 2 h at 65 °C, and then released from the master mold. Thermo-curable epoxy resin was poured on the PDMS mold, and cured in two steps of 125 °C and 165 °C for 90 min and 60 min, respectively. This epoxy substrate was released from the PDMS mold and UV photolithography was used to pattern the substrate for metal deposition. Typically, electron beam evaporator was used to deposit metals (Ti 200 Å/Au 500 Å/Pt 500 Å/Au 2000 Å). Additionally, DC sputtering and electrochemical gold deposition was used to fabricate gold electrode chips to study the effect of different metal layers in solution. After metal deposition, the device was passivated in photoresist and, using photolithography, two open areas were defined on the gold electrode. One sensing electrode was connected to the gate region of MOSFET, and the other to the reference electrode where gate bias was applied (Figure 1). The area of the sensing electrode was 600 μm × 600 μm, and the gap between the two open areas was 65 μm. 

### 2.2. Surface Functionalization

The surface functionalization was carried out by immobilizing single-stranded DNA probe molecules onto the gold surface of the sensor chip. Firstly, thiolated single-stranded DNA (thiol group at 5-prime side) was reduced using Tris(2-carboxyethyl) phosphine (TCEP) (SIGMA-ALORICH, St. Louis, MO, USA) by mixing the aptamer and the reducing agent in the molar ratio 1:1000 in tris-ethylenediaminetetraacetic acid (EDTA) buffer (TE buffer). After incubation in room temperature for 15 min, the aptamer solution was heated to 95 °C to disrupt the hydrogen bonds and yield an uncoiled ssDNA structure, and later flash cooled and dropped on the gold electrodes on the sensor for incubation at room temperature for 24 h.

### 2.3. Test Sample Preparation

Different test environments were prepared for investigations. The test buffer used in all the experiments was 1× PBS (with 137 mM NaCl) (AMRESCO, Inc., Cuyahoga, OH, USA). To adjust the viscosity of the test medium, PBST and glycerol solutions were prepared. 10% PBST was prepared by mixing tween 20 in 1× PBS. Glycerol solutions of 10%, 50% and 100% were prepared in 1× PBS. The sample delivery onto the sensor was carried out by pipette dropping 100 µL of the test solution onto the sensing area. The electrical measurements were conducted under static conditions of liquid droplet.

### 2.4. Field Effect Transistor (MOSFET)

In this study, the signal transduction was carried out using n-channel depletion mode MOSFET (#LND150; Supertex Inc., Sunnyvale, CA, USA)The gate voltage was applied in the form of a short duration (100 µs) pulsed bias with an amplitude of 1 V, and the drain voltage was a constant bias of 2 V. The difference of the absolute drain current before and after applying the gate voltage was used as the sensor signal and was denoted as gain.

### 2.5. Statistical Methods for Analysis of Stability

To account for variations arising from MOSFET drain current drift due to thermal or external noise, we subtracted the gain of MOSFET from the absolute gain obtained as output from our sensor. This difference, denoted as ΔGAIN, was therefore used to analyze all the experimental results in this work. We defined fluctuation of the electrical signal as sensor stability, and used the standard deviation formula to calculate the electrical signal fluctuation. However, one sensor’s ΔGAIN may be different from that of another, due to sensor-to-sensor variations. In order to statistically analyze all the sensors’ standard deviations, we calculated the average of ΔGAIN for each sensor, as shown in Figure 2a, and shifted the average of ΔGAIN to the same baseline value, as shown in Figure 2b. Finally, the baseline adjusted sensor responses were plotted with respect of time, as shown in Figure 2c. We then can compare sensors in different solutions through time, as shown in Figure 2d.

## 3. Results and Discussion

### 3.1. Sensor Design and Implementation

The sensor structure consisted of an FET and a reference electrode supplied with a pulsed gate voltage. The schematic diagram of the high field modulated extended gate FET biosensor is shown in Figure 1. Two gold electrodes separated by a narrow gap of 65 µm formed the sensing region on a detachable sensor chip (Figure 1a). Several pairs of gold electrodes were fabricated on a single sensor chip, forming a sensor array. In each pair of electrodes, one was connected to gate voltage supply and the other was connected to the gate terminal of the MOSFET used for signal transduction. The electrode connected to the gate bias formed the reference electrode through which bias waas applied to the test solution. The electrode connected to the gate terminal of MOSFET was the extended gate metal, which received the voltage drop across the solution, thereby modulating the voltage drop in the MOSFET dielectric and hence the channel current. 

By placing the reference electrode and the extended gate metal within a short distance of 65 µm, we essentially created a high field across the test solution, when gate pulse bias was applied.

The principle of our high field modulated extended gate FET biosensor is detailed in our previous works [17,24]. Briefly, when we apply pulsed gate voltage to the reference electrode (with the source drain of MOSFET being supplied with a DC bias), the voltage drops in the test liquid placed in the gap and through the extended gate metal, reaches the FET dielectric where it drops, and further modulates the FET drain current. The sensing structure simulates a liquid capacitor type model. The electrical double layers at the solid/liquid interfaces generate a solution capacitance, which can be viewed as an additional dielectric to the FET sensor system apart from the FET’s gate dielectric. Therefore, changes in the solution capacitance owing to changes in test solution itself or interfacial interactions at the solid/liquid boundaries lead to different potential drops in the FET gate dielectric, and hence the modulation of the transistor drain current. The schematic illustration of the extended gate FET biosensor is depicted in Figure 1b, and the real view image of the prototype system is shown in Figure 1c. The pulsed voltage applied as gate voltage is schematically depicted in Figure 1d. The pulse duration is 100 µs, with an amplitude of 1 V. Typical drain current response obtained for the pulsed gate bias application is depicted in Figure 1e. 

### 3.2. Influence of Sensor Surface Characteristics

FET biosensors are surface affinity sensors. The aqueous sample being tested is placed on the sensor surface, and the physical/chemical/biological interaction of the components of the liquid with the sensor surface is transduced into electrical signals by the FET. Prior to investigating the interaction of liquid samples with the sensor surface bound receptors, we need to understand the effects of different types of sensor surfaces that contribute to different electrical signals. To conduct this study, we fabricated the gold electrodes on the sensor chips with different techniques. We used electroplating, DC sputtering, and e-beam evaporation to deposit gold metals on the sensor chip. The differences in the gold electrodes deposited using these three methods primarily occur in terms of their surface topography [25,26,27]. Therefore, we measured the surface roughness of the metals deposited via the three methods, using atomic force microscope (AFM), the results of which are depicted in Figure 3. The electroplated gold electrodes had greater surface roughness (68.251 nm), while sputtered (19.49 nm) and evaporated gold electrodes (20.498 nm) had comparatively smoother surface profiles, i.e., less surface roughness. 

To study the electrical signals from these metal electrodes, we dropped 50 µL of buffer (1× PBS with 137 mM NaCl) on the sensor chip, covering the gold electrodes completely, and applied a gate pulse (1 V; 100 µs on-time) for every 10 min and recorded the FET drain current up to 60 min. At every 10 min, the bias was applied for 1 time unit, acquiring the electrical data 10 times. The standard deviation at every point of sensor measurement indicates the variation of drain current of MOSFET arising from more than 70 separate bias applications. Therefore, the standard deviation is an indicator of the surface potential/charge density/solution capacitance generated by the pulse application. The standard deviation thus relays a measure of stability of the sensor. The results are shown in Figure 4a–c, depicting the electrical characteristics of electroplated, sputtered, and evaporated metal electrodes, respectively. From analyzing the results, we can see that, in the test environment of 1× PBS, the standard deviation was larger for the electroplated chip compared to the sputtered and evaporated chips. This means that there is a comparatively large fluctuation of surface potential for the electroplated metal, and hence lower stability. This can be reasoned from the surface roughness described previously. The electrostatic interaction of the ionic species in the aqueous sample and the metal layer is dependent on the surface topography. Greater surface roughness contributes to larger fluctuations in potential and larger instability. From Figure 4b,c, we can see that sputtered and evaporated metals give rise to very similar standard deviation values in a 1× PBS test environment and, incidentally, their surface roughness values were also comparable and smaller than that of electroplated gold. Thus, the stability was improved in sputtered and evaporated metal electrode chips. Another plausible reason for the decreased stability of electroplated chips is the presence of easily oxidized metals on the surface of the metal electrodes. Electroplated gold electrodes mainly contain copper, which can oxidize in aqueous environments and diffuse into the test sample. This would result in different charge densities or surface potentials at different times, which we are observing as larger standard deviation values in Figure 4. 

### 3.3. Influence of Solution Composition

The stability of sensor characteristics is not solely dependent on the surface morphology of the gold electrodes. It is also dependent on the composition of the aqueous solution, which contains different species (both charged and uncharged) that closely interact with the sensor surface and exert weak forces that affect the overall surface potential. We analyzed the effects of metal–solution interaction by varying temperature and viscosity of the aqueous samples. Figure 4 demonstrates the electrical results obtained while testing electroplated, sputtered, and evaporated metals in different environments. Figure 4a–c shows the drain current signal in different temperatures, i.e., 1× PBS test medium at room temperature and 3 °C, for electroplated, sputtered, and evaporated metal chips. It is obviously seen that the lower temperature environment resulted in lower standard deviation values, which means that the stability was improved considerably in decreased temperatures. Additionally, to investigate the effect of viscosity, we prepared different solutions: 10% PBST and glycerol in varying concentrations (10%, 50%, 90%, and 100%). The viscosities of the different solutions were: 1× PBS < 10% PBST < 10% glycerol < 50% glycerol < 100% glycerol. By comparing sub-sections (a), (b), and (c) of Figure 4, we can see that the standard deviation values decreased with increasing viscosity of the test medium (measured at room temperature), except for 100% glycerol. In the three different metal electrode chips, 100% glycerol consistently resulted in higher standard deviations, meaning larger instability. This is because 100% glycerol is a polyol found in all lipids. The test environment does not contain any ionic species, and, when gate pulse is applied, the weak polarization of glycerol creates large impedance or more potential drop in the test liquid compared to 1× PBS, which contains ions that readily form EDL at the solid/liquid interfaces. The interfacial potential in the case of 100% glycerol is therefore dependent on the polarizability of the molecule itself. On the other hand, 10%, 50%, and 90% glycerol compositions were prepared in 1× PBS medium and therefore contained ions that generate EDL, leading to much lower impedance and more potential drop in the FET dielectric rather than in the test medium. The interfacial potential will be dominantly determined by the charge density from the double layers. 

Figure 4d describes the standard deviation values obtained for different types of metal electrodes under different conditions of test environment. Table 1 summarizes the electrical characteristics in terms of test environment (temperature and viscosity) and surface topography (electroplated, sputtered, and evaporated). The table can be used to quantitatively analyze the effect of solution and metal surface on the stability of the sensor.

To comprehend the electrical characteristics, we needed to analyze the physical phenomenon that occurs at the interfaces and bulk of the solution. The particles in the solution constantly interact with each other and the sensor surface. The mobility of a particle in fluid can be studied using the Stokes equation, which describes mobility (*µ*) as a function of particle radius (*R*) and viscosity (*η*):(1)$$\mu =\frac{1}{6\Pi \eta R}$$

Therefore, greater viscosity of the fluid will result in lower mobility of the particles and thus, fewer random collisions. 

Considering Einstein’s relation, which relates the diffusion coefficient (*D*) to the mobility (*µ*) and temperature (*T*) as:(2)$$D=\mu k_{B}T$$

Combining Equations (1) and (2), we obtain the Stokes–Einstein equation:(3)$$D=\frac{k_{B}T}{6\Pi \eta R}$$

This means that at higher temperatures and lower viscosities of fluids, the diffusivity will be larger. Diffusion is facilitated by Brownian motion [28,29], which is the random motion of particles in a medium. Brownian motion induces fluctuation of drain current due to variations in the interfacial potential resulting from randomly colliding particles. Inversely, when diffusivity is less at conditions of lower temperature and higher viscosity, less fluctuation and hence greater sensor stability can be observed, as demonstrated in our experiments. 

### 3.4. Influence of Surface Functionalization

The signal transduction of FET biosensor relies on the surface bound receptor and ligand interactions. The change in surface potential and the charge density due to the electrostatic binding of receptor and target biomolecule lead to modulation of the FET drain current. Therefore, surface functionalization is an important aspect of surface affinity sensors such as FET biosensors. To investigate the effect of sensor stability and surface functionalization, we characterized different sensor chips fabricated with evaporated gold electrodes, with varying levels of functionalization. The gold electrodes were cleaned using UV ozone etching, followed by washing in mild acid and deionized water. The cleaned sensor chips were placed in testing buffer and electrical measurement was carried out for 60 min, at intervals of 10 min. The same electrical measurement process was also repeated after the sensor chips were functionalized with aptamer molecules (single stranded DNA) in final concentrations of 1, 5, and 10 µM. The results are shown in Figure 5. When there were no surface bound aptamer molecules (Figure 5a), the standard deviation was the largest. To verify the immobilization of the aptamer, we utilized fluorophore tagged aptamer molecules to be immobilized on the sensor surface. The fluorescence images depicted alongside the electrical results demonstrate the level of surface functionalization. In the functionalized sensor chips with varying densities of immobilized aptamers, the standard deviation values were proportional to the final concentration of immobilized aptamer: 1 µM > 5 µM > 10 µM. The corresponding fluorescence images are shown alongside Figure 5b–d. It can be seen that when the final concentration of aptamer used for immobilization was 10 µM (the highest), the fluorescence intensity was the highest and the electrical stability was significantly improved. Conversely, when the final aptamer concentration of 1 µM (lowest) was used, the intensity was the lowest and electrical stability was poor, better only than the blank condition. The fluorescence intensity analysis with respect to final concentration of aptamers used for immobilization is depicted in Figure 5e. Table 2 summarizes the standard deviation values obtained under different conditions of surface functionalization. From the results in Figure 5, it can be deduced that the presence of surface bound aptamer molecules improves the electrical stability of the sensor. This can be attributed to the addition of charged or polarizable molecules on to the sensor surface, which re-distribute the electrical double layer in the gold/test liquid interface. The charge density in the interface is influenced and directed by the immobilized aptamers, which counteract the random fluctuations created by Brownian motion of particles. Addition of charged molecules to the gold surface has previously been shown to significantly influence the interfacial potential. In our experiments, we demonstrated that the control of surface topography, solution composition and consistency, ambient conditions, and surface functionalization can regulate the interfacial interactions leading to improvements in the electrical stability of extended gate FET biosensors.

Furthermore, we conducted experiments to detect miRNA targets in 1× PBS environments with the devices immobilized with different concentrations of probe (aptamer): 1, 5, and 10 µM. After probe immobilization, 15 µL of target miRNA (miR-21) was dropped on the sensor surface, and the device was heated to 43 °C and maintained at that temperature for 5 min to eliminate mismatched RNA sequence binding. Following this, the device was cooled down to room temperature in 3 min and electrical measurements were carried out. The results are depicted in Figure 6a. As the concentration of miR-21 increased, the current gain increased. For the device with 1 µM of probe concentration, there was a clear saturation of sensor signal after 100 fM of miR-21 concentration (Figure 6a), however, for the devices with 5 and 10 µM of probe concentrations, the sensor dynamic range was wider (0–1 pM [miR-21]) and sensor signal did not saturate at higher target miRNA concentrations. This can be related to the availability of more binding sites for the target miRNA on the sensor surface, which allows the sensor to have wider dynamic range, as in the case of devices with 5 and 10 µM probe concentrations. The increased stability in higher probe concentration may be attributed to the size and surface coverage of the immobilized aptamers. To verify the dependence of sensor signal on surface coverage, we tested 1× PBS containing miRNA on bare gold electrodes, i.e., non-functionalized sensor electrodes. The results are shown in Figure 6b. It can be seen that the higher concentrations of miRNA that were dropped on the bare gold surface of the sensor caused a slight decrease in sensor signal (from miRNA concentrations of 100–1000 fM). This could explain the saturation characteristics observed in Figure 6a for the condition of 1 µM aptamer concentration: the surface coverage was comparatively lesser (than 5 and 10 µM aptamer concentrations), and therefore non-specific binding of miRNA at higher concentrations may be more pronounced, leading to lowering of sensor signal after it has reached saturation. Furthermore, the DNA probe’s size is much larger compared to ions, and, when immobilized on the sensor surface, their height is greater than the double layer at the interface. Unlike ions, they are less prone to Brownian-motion-induced fluctuations, and thus they stabilize the gold surface. More surface coverage of the immobilized probe molecules improved the stability. Therefore, by optimizing the immobilized probe concentration on the sensor surface, we can enhance the stability and the sensitivity of the FET biosensor, which is quite important for clinical applications that require testing of human plasma or whole blood samples. These results show that our sensor has the potential to be used for early diagnosis and screening for various types of cancers that have identified miRNA as biomarkers. 

## 4. Conclusions

This research conducted investigations into the factors that influence the electrical stability of high field modulated extended gate FET biosensors in aqueous solutions. The test results demonstrate that surface topography of the gold electrodes influences the drain current variations. The metal electrodes deposited using sputtering and e-beam evaporator yielded the lowest noise in their signal. The temperature and viscosity of the test liquid also contributed towards electrical instability, in agreement with the Stokes–Einstein equation that explains Brownian motion of particles. Reducing the temperature and increasing the viscosity of the test sample decreased the fluctuation in electrical signals. The surface modification with charged molecules such as aptamers lead to improved sensor stability, due to the charge re-distribution in the electrical double layers at the solid/liquid interfaces. The extended gate FET biosensor can be optimized for improved sensor performance in aqueous media. It was also demonstrated that, by optimizing the functionalization of FET sensor, we can tune the stability as well as the sensitivity of miRNA detection to realize a biosensor that is suitable for clinical applications where miRNA is used as biomarkers of diseases. 

## Figures and Tables

**Figure 1 sensors-19-01484-f001:**
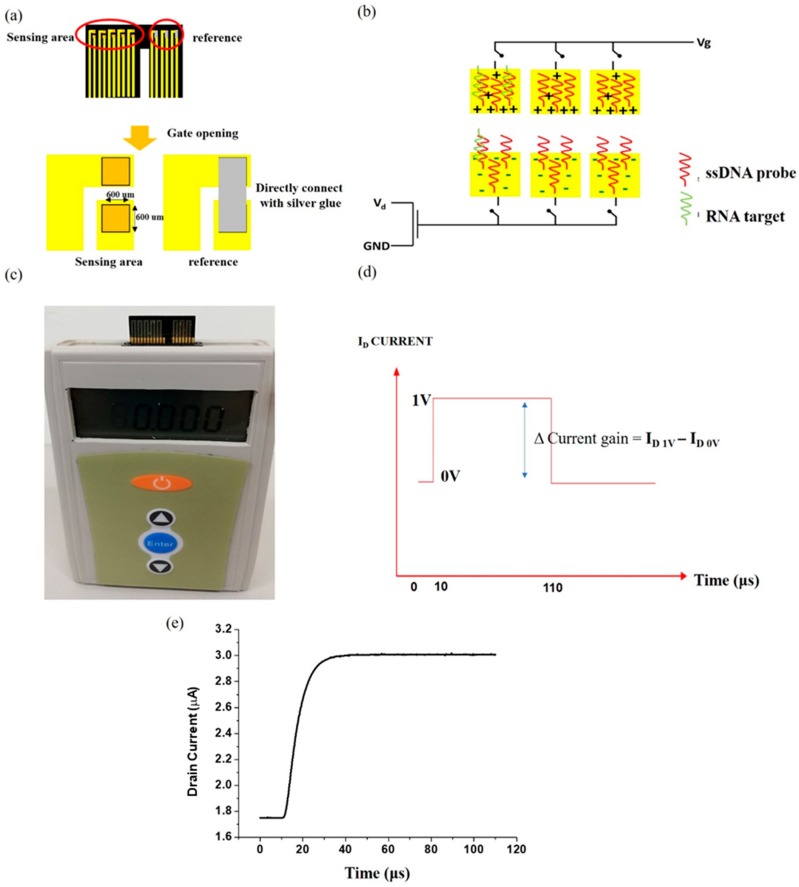
(**a**) Sensing area and reference electrode opening of extended gate chip. (**b**) Schematic of functionalized high field modulated extended gate field effect transistor (FET) biosensor array. (**c**) Real image of high field modulated extended gate FET biosensor. (**d**) Pulse bias applied as gate voltage to the FET sensor. (**e**) Typical drain current versus time curve of FET biosensor measured in 1× PBS environment.

**Figure 2 sensors-19-01484-f002:**
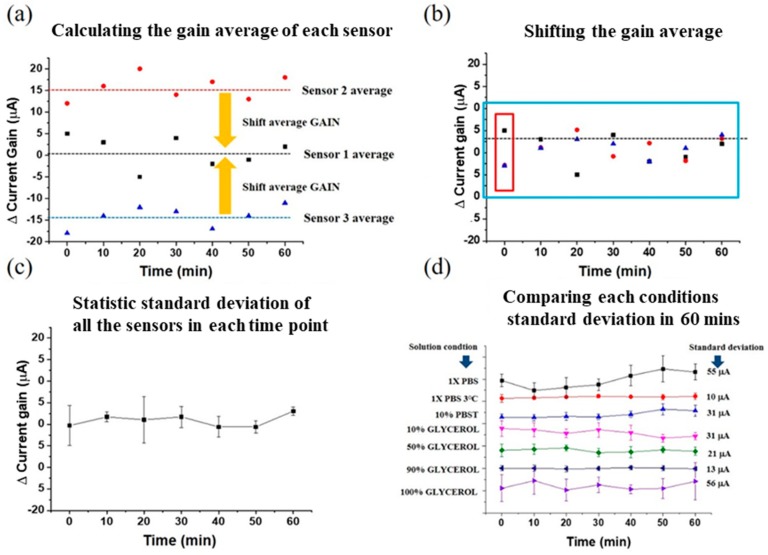
(**a**) Average gain calculation of individual sensors. (**b**) Shifting the gain average to the same baseline value. (**c**) Baseline shifted sensor response with respect to time. (**d**) Comparison of sensors in different solutions through time.

**Figure 3 sensors-19-01484-f003:**
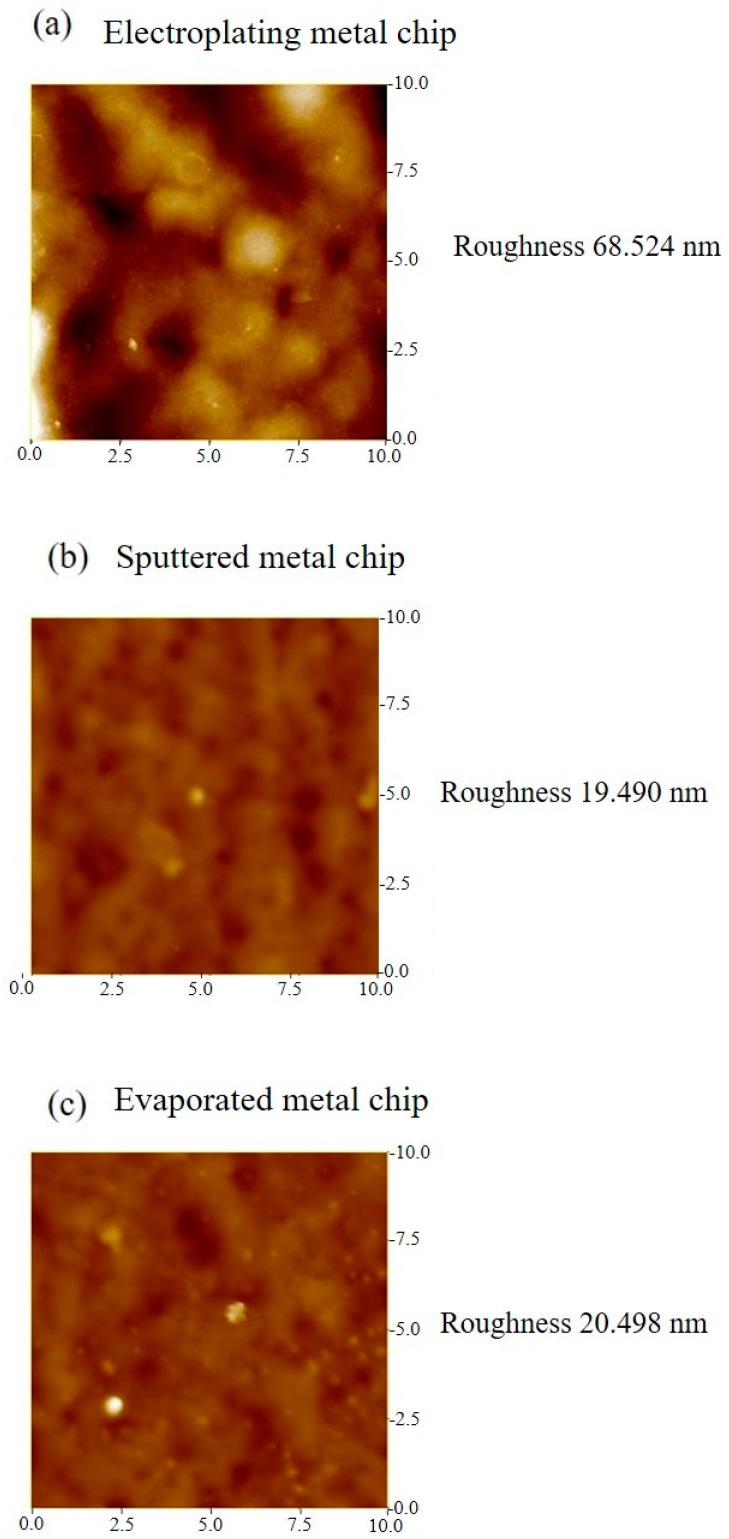
Surface roughness measurement by atomic force microscope (AFM) for (**a**) electroplated metal chip, (**b**) sputtered metal chip, and (**c**) evaporated metal chip.

**Figure 4 sensors-19-01484-f004:**
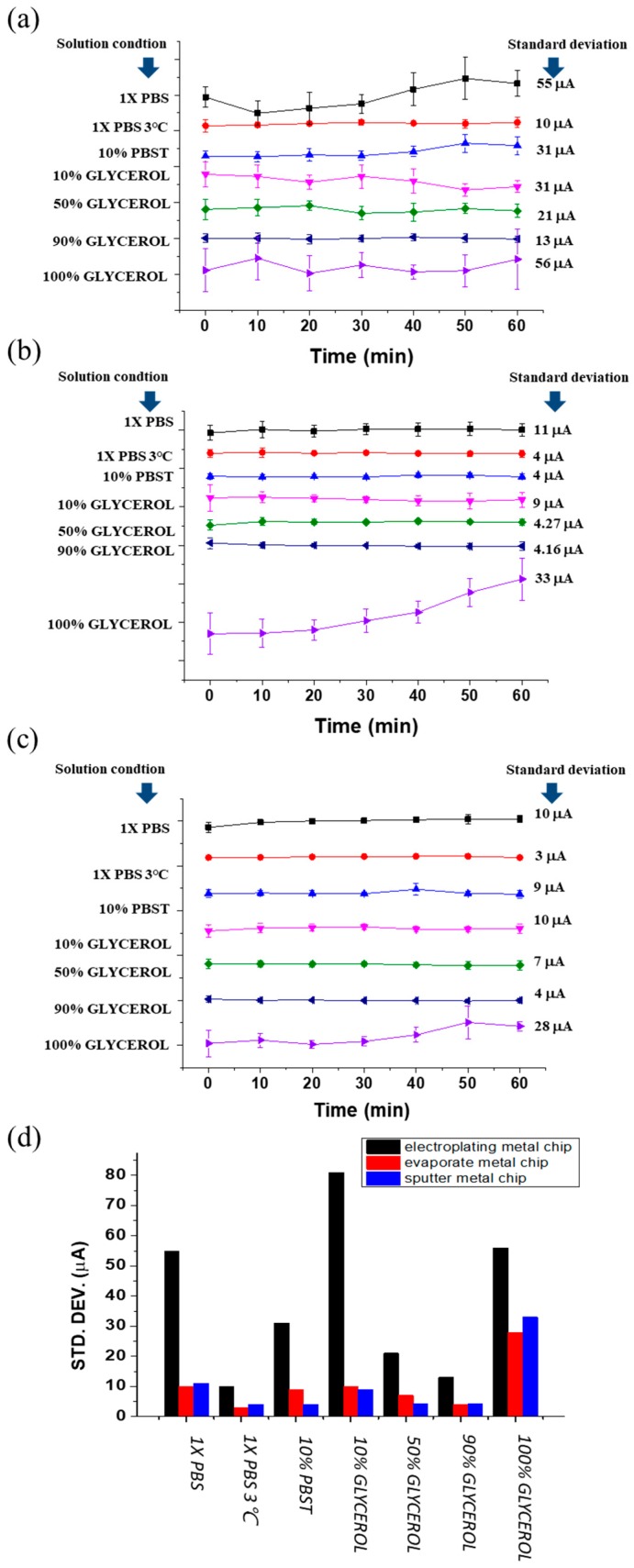
(**a**) Electroplated gold electrode chip stability in different conditions (1× PBS in RT, 1× PBS in 3 °C, 10% PBST, 10/50/90/100% glycerol in 1× PBS). (**b**) Sputtered gold electrode chip stability in different conditions (1× PBS in RT, 1× PBS in 3 °C, 10% PBST, 10/50/90/100% glycerol in 1× PBS). (**c**) Evaporated gold electrode chip stability in different conditions (1× PBS in RT, 1× PBS in 3 °C, 10% PBST, 10/50/90/100% glycerol in 1× PBS). (**d**) Standard deviation versus test environment.

**Figure 5 sensors-19-01484-f005:**
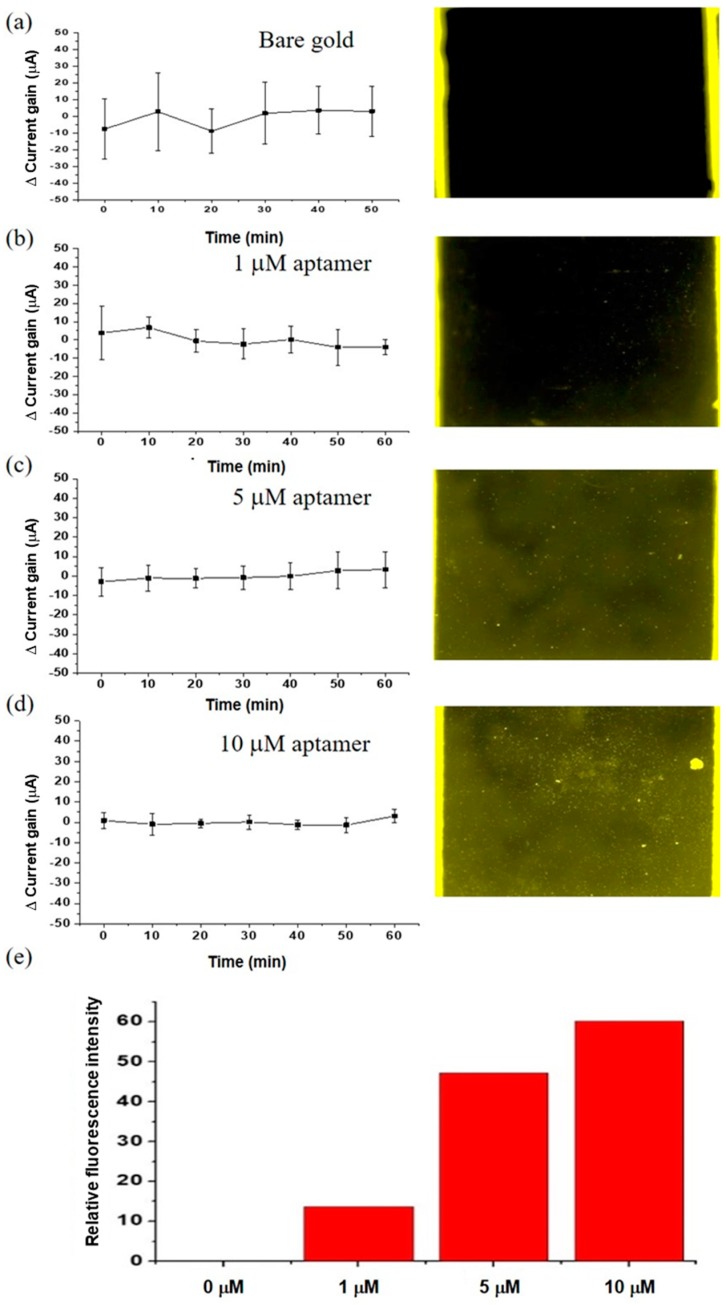
Sensor signal stability in functionalized sensor chips with final aptamer concentrations of (**a**) 0 µM (blank), (**b**) 1 µM, (**c**) 5 µM, and (**d**) 10 µM. Surface functionalizations carried out with final aptamer concentrations of 0, 1, 5, and 10 µM are depicted in Figures (**a**) through (**d**), respectively. (**e**) Fluorescence intensity of functionalized sensor chips with final aptamer concentrations of 0, 1, 5, and 10 µM.

**Figure 6 sensors-19-01484-f006:**
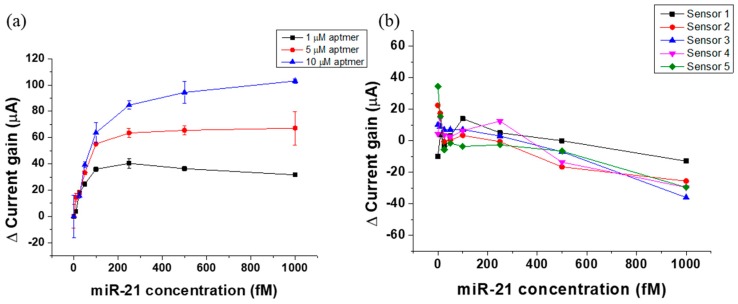
(**a**) 1/5/10 µM DNA probe immobilized sensor chips calibration curve showing current gain versus miRNA concentration (miRNA concentration 0/10/25/50/100/250/500/1000 fM). (**b**) Testing miRNA (0/10/25/50/100/250/500/1000 fM) in 1× PBS on bare gold surface of sensor (non-functionalized sensor).

**Table 1 sensors-19-01484-t001:** Standard deviation values corresponding to different test environments for three different types of metal depositions (electroplated gold electrode chip, evaporated gold electrode chip, sputtered gold electrode chip) (N = number of measurements).

Viscosity (m Pa.s)	Test Condition	Electroplated Gold Electrode Chip Std. Dev.	Evaporated Gold Electrode Chip Std. Dev.	Sputtered Gold Electrode Chip Std. Dev.
1.60	RT 1× PBS	55 μA N = 77	10 μA N = 70	11 μA N = 84
NA	3 °C 1× PBS	10 μA N = 84	3 μA N = 84	4 μA N = 63
1.79	10% PBST	31 μA N = 98	9 μA N=105	4 μA N = 56
1.83	10% GLYCEROL	31 μA N = 77	10 μA N = 84	9 μA N = 63
6.83	50% GLYCEROL	21 μA N = 77	7 μA N = 84	4.27 μA N = 70
262.4	90% GLYCEROL	13 μA N = 84	4 μA N = 70	4.16 μA N = 70
782	100% GLYCEROL (2.96)	56 μA N = 84	28 μA N = 91	33 μA N = 63

**Table 2 sensors-19-01484-t002:** Aptamer concentrations and corresponding standard deviation values expressed in μA. (N = number of measurements).

Aptamer Concentration	Standard Deviation (μA)
Blank (N = 70)	17
1 µM (N = 74)	9.3
5 µM (N = 77)	7.1
10 µM (N = 104)	3.7

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
