# Peer review of "Investigation of Electrical Stability and Sensitivity of Electric Double Layer Gated Field-Effect Transistors (FETs) for miRNA Detection"

_sensors, 2019, doi:10.3390/s19071484_

Round 1
Reviewer 1 Report
Reliable stability is one of the biggest issues in bio-sensors, though it is not studied and reported enough to be commercialized. This article reports the stability of the electric double layer FET based miRNA sensor. The effect of surface morphology of the sensing area, temperature, viscosity, and surface functionalization on the sensing stability has been investigated and well explained. Also, small LOD of this sensor with wide dynamic detection range is quite impressive. The experimental approach is systematic and the conclusion is reasonable. I recommend the publication of this paper, and expect this article will attract many interest of the Sensors readers.
Author Response
Responses included in the attached file.

Reviewer 2 Report
The article is a well-written and interpretation of results is clearly presented. So, I recommend to publish the article, however, minor revisions are needed.
- The abstract and the conclusions are very general. It is recommended to be more specific and addressable. E.g. which was the best sensor and why? What were the main findings, etc. You could also mention the two temperature points (RT and 3°C), because now from abstract it seems that wide temperature range was used.
- Tables 1 and 2 are not clear without reading the whole article. Tables must be easily understandable at first sight. E.g. description of Tab.1 (“Table 1. Summary of electrical characterization”) must be improved. What does mean N=77 in the table 1? What does mean electroplated (mA), evaporated (mA), sputtered (mA)?, etc. So, without reading the whole article we cannot know.
- The same for Table 2. Instead of functionalization, it is better to use „aptamer concentration“ or something similar.
- The „current gain“ on the y-axis in all figures is confusing. E.g. see Fig.2, where Fig.2a,b,c,d have the same label („current gain“), but they are differ. Because as I understand, the Fig. 2a is the Dcurrent gain (i.e. absolute (or measured) current gain subtracted by the current gain of the MOSFET). Further, the Fig. 2b, is something else. It should be e.g. normalized current gain (ie. Dcurrent gains shifted to 0). Fig 2c is the same as Fig. 2b, but there is Fig. 2d, where the measured data are in the interval 300-800 mA. What kind of current gains are in the Fig. 2d? Are they absolute (measured data) or Dcurrent gains (i.e. absolute (measured) current gain subtracted by the current gain of the MOSFET)? So, it is confusing if all figures have the same y-label, but the data are different. Please, make it more understandable. It is the same comment also for Fig. 4, Fig. 5 and Fig.6., the label for y-axis of current gains must be clearly defined.
- In the Fig.5, the label of y-axis “Relativity Florence Intensitivity” must be changed to “Relative fluorescence intensity”.
Author Response

(The authors gave the same response as above.)

Reviewer 3 Report
The paper studies some technological aspects of FETs biosensors. The influence of surface roughness, temperature, viscosity and surface modification on the sensor stability is investigated. The study addresses important yet underappreciated topics that should deserve more attention. Before publication, several concerns should be addressed:
· The surface roughness is not investigated carefully. There may be differences in chemical compositions between different techniques. The cleanliness and purity of the gold may depend on the gold deposition technique. The authors should do some XPS analysis or similar to make sure that the surfaces are chemically the same. Otherwise it is not possible to attribute the changes to roughness only.
· The experiments with glycerol are not convincing because the authors change both the viscosity and the ionic strength, i.e. at least two parameters at the same time. In my opinion, it is necessary to study the effect of ionic strength independently, i.e. without glycerol addition, to clarify whether the observed effect is only due to viscosity.
· The authors do not comment on how the sample was delivered to the sensor and whether the experiments were performed under flow conditions or not. Electrokinetic phenomena such as flow potential can strongly influence the sensor stability as well and should be experimentally studied.
· The authors claim that the Debye screening limitations can be overcome using this approach. However, there are also other methods to achieve this that are completely ignored in the introduction, e.g.:
o 10.1021/acs.nanolett.5b00133
o 10.1002/adma.201403541
o 10.1002/admt.201800186
o 10.1021/acssensors.7b00187
Overall, I recommend publication if the above mentioned concerns can be addressed.
Author Response

(The authors gave the same response as above.)
